# Epigenetic Dysregulation and Its Correlation with the Steroidogenic Machinery Impacting Breast Pathogenesis: Data Mining and Molecular Insights into Therapeutics

**DOI:** 10.3390/ijms242216488

**Published:** 2023-11-18

**Authors:** Pulak R. Manna, Shengping Yang, P. Hemachandra Reddy

**Affiliations:** 1Department of Internal Medicine, School of Medicine, Texas Tech University Health Sciences Center, Lubbock, TX 79430, USA; hemachandra.reddy@ttuhsc.edu; 2Department of Biostatistics, Pennington Biomedical Research Center, Louisiana State University, Baton Rouge, LA 70808, USA; shengping.yang@pbrc.edu; 3Neurology, Departments of School of Medicine, Texas Tech University Health Sciences Center, Lubbock, TX 79430, USA; 4Public Health Department of Graduate School of Biomedical Sciences, Texas Tech University Health Sciences Center, Lubbock, TX 79430, USA; 5Department of Speech, Language and Hearing Sciences, School Health Professions, Texas Tech University Health Sciences Center, Lubbock, TX 79430, USA; 6Department of Pharmacology and Neuroscience, Texas Tech University Health Sciences Center, Lubbock, TX 79430, USA

**Keywords:** breast cancer, estrogen/E2, epigenetic enzymes, STAR, steroidogenic machinery, hormone receptors, HDAC inhibitors, therapeutics

## Abstract

Breast cancer (BC) is a heterogeneous condition and comprises molecularly distinct subtypes. An imbalance in the levels of epigenetic histone deacetylases (HDACs), modulating estrogen accumulation, especially 17β-estradiol (E2), promotes breast tumorigenesis. In the present study, analyses of The Cancer Genome Atlas (TCGA) pan-cancer normalized RNA-Seq datasets revealed the dysregulation of 16 epigenetic enzymes (among a total of 18 members) in luminal BC subtypes, in comparison to their non-cancerous counterparts. Explicitly, genomic profiling of these epigenetic enzymes displayed increases in HDAC1, 2, 8, 10, 11, and Sirtuins (SIRTs) 6 and 7, and decreases in HDAC4–7, –9, and SIRT1–4 levels, respectively, in TCGA breast tumors. Kaplan–Meier plot analyses showed that these HDACs, with the exception of HDAC2 and SIRT2, were not correlated with the overall survival of BC patients. Additionally, disruption of the epigenetic signaling in TCGA BC subtypes, as assessed using both heatmaps and boxplots, was associated with the genomic expression of factors that are instrumental for cholesterol trafficking/utilization for accelerating estrogen/E2 levels, in which steroidogenic acute regulatory protein (STAR) mediates the rate-limiting step in steroid biosynthesis. TCGA breast samples showed diverse expression patterns of a variety of key steroidogenic markers and hormone receptors, including *LIPE, CYP27A1, STAR, STARD3, CYP11A1, CYP19A1, ER, PGR,* and *ERBB2*. Moreover, regulation of STAR-governed steroidogenic machinery was found to be influenced by various transcription factors, i.e., *CREB1, CREM, SF1, NR4A1, CEBPB, SREBF1, SREBF2, SP1, FOS, JUN, NR0B1,* and *YY1*. Along these lines, ingenuity pathway analysis (IPA) recognized a number of new targets and downstream effectors influencing BCs. Of note, genomic, epigenomic, transcriptional, and hormonal anomalies observed in human primary breast tumors were qualitatively similar in pertinent BC cell lines. These findings identify the functional correlation between dysregulated epigenetic enzymes and estrogen/E2 accumulation in human breast tumors, providing the molecular insights into more targeted therapeutic approaches involving the inhibition of HDACs for combating this life-threatening disease.

## 1. Introduction

Breast cancer (BC) is the most prevalent malignant disorder in women, in which certain BC subtypes are aggressive and resistant to drugs, and it is the second greatest cause of cancer-related death among women worldwide [1]. Dysregulation of a variety of processes and factors, including epigenetic alterations, plays a crucial role in the pathogenesis and progression of BCs [2,3,4]. Histone deacetylases (HDACs) are a family of epigenetic enzymes that remove the acetyl group on histone proteins as well as lysine residues on non-histone proteins, resulting in chromatin remodeling, regulation of the transcriptional machinery, and post-translational modifications (PTMs) of non-histone proteins, involving genomic stability [5,6,7]. The mammalian HDAC family consists of 18 members that are grouped into four classes in which Class I comprises HDAC1, 2, 3, and 8; Class II includes HDAC4, 5, 6, 7, 9, and 10; Class III possesses Sirtuin (SIRT) 1–7, and Class IV contains HDAC 11, which are frequently dysregulated in BCs [6,8].

Breast pathogenesis is influenced by genomic and epigenomic alterations, resulting in tumor initiation, progression, and heterogeneity by disrupting the equilibrium between oncogenes and tumor suppressor genes. BCs are categorized into four molecular subtypes: (i) luminal A, estrogen-receptor-positive (ER+), especially ERα, progesterone receptor (PR+), and human epidermal growth factor receptor-negative (Her−, also called ERBB2); (ii) luminal B, ER+, PR+, and Her+; (iii) Her2/ERBB2, ER−, PR−, and Her+; and (iv) basal-like, ER−, PR− and Her− (also termed as triple negative, TNBCs), [9,10,11,12,13]. The majority of BC subtypes (≥80% of all cases) are hormone-sensitive and express ERα, PR, and/or Her/ERBB2, and the remaining ~15% are TNBCs and do not express those three receptors. It is unambiguous that ER+/PR+ BC is primarily activated by estrogens, particularly E2, that are synthesized from androgens using the aromatase enzyme. However, aromatase (*CYP19A1*) has been shown to be indiscriminately expressed in both malignant and non-malignant breast tissues [14,15,16]. In accordance, aromatase inhibitors (AIs) have been frequently used for BC treatment in post-menopausal women; however, AIs generate undesirable side effects, including the AI resistance that recurrently causes cancer death, warranting an improved therapy for this devastating disease. Since epigenetic enzymes (HDACs) are frequently dysregulated and/or mutated, attention has been placed upon the development of HDAC inhibitors (HDACIs) for BC therapies [9,13,15,16]. HDACIs are clinically efficacious and safe and display limited toxicity (in comparison to AIs) against multiple oncogenic events, in which the inhibition of HDACs results in the acetylation of numerous histone and non-histone substrates, including tumor suppressor proteins and oncogenes [12,13,16,17,18,19,20]. Acetylation is pivotal in protein expression and function and this post-translational modification (PTM) is particularly influenced by histone acetyl transferases (HATs) and HDACs, which could modulate estrogen/E2 biosynthesis [13,16].

Steroid hormones, including E2, are made from cholesterol, and malfunction in the steroidogenic pathway affects a variety of factors/processes that influence biochemical steps, ranging from cholesterol trafficking/availability to estrogen/E2 accumulation. These include HSL/*LIPE* (hormone-sensitive lipase), *CYP27A1* (cytochrome P450 oxidase, 27-hydroxylase), STAR (steroidogenic acute regulatory protein, also called STARD1, STAR-related lipid transfer domain 1), STARD3, *CYP11A1* (cytochrome P450 family 11 subfamily A member 1), *HSD3B1* (3β-hydroxysteroid dehydrogenase/δ(5)-δ(4)isomerase type I), *HSD17B1* (17β-hydroxysteroid dehydrogenase 1), *CYP19A1* (aromatase), *ESR1* (estrogen receptor 1, ERα), *ESR2* (estrogen receptor 2, ERβ), *PGR* (progesterone receptor), and Her2/ERBB2 (human epidermal growth factor receptor 2/the erythroblastosis oncogene-B2). Among these factors, the STAR protein predominantly regulates steroid hormone biosynthesis via mechanisms that enhance the transcription, translation, or activity of this cholesterol transporter in a variety of tissues. Studies have shown that trans-regulation of the STAR gene is finely tuned by various positive and negative factors, including CREB1 (cAMP responsive element binding protein 1), CREM (CAMP-responsive element modulator), SF1 (steroidogenic factor 1), NR4A1 (nuclear receptor 4A1, also called Nur77), CEBPB (CCAAT/enhancer-binding protein-β), *SREBF1* (sterol regulatory element-binding transcription factor 1, also called SREBP1), *SREBF2* (sterol regulatory element-binding transcription factor 2, SREBP2), SP1 (specialty protein 1), FOS (Fos proto-oncogene), JUN (Jun proto-oncogene), NR0B1 (nuclear receptor subfamily 0 group B member 1), and YY1 (yin yang 1), and a balance between the inducer and repressor functions of these factors presumably allows for a fine-tuning of the steroidogenic machinery [21,22,23].

TCGA RNA-Seq datasets offer a comprehensive understanding of the molecular basis of BC subtypes using high-throughput genome sequencing and bioinformatics, and, accordingly, their prognosis and therapeutic interventions [9,11,24]. Moreover, to examine the efficacy and sensitivity of experimental agents/drugs, pertinent BC cell lines are excellent tools for in-depth understanding of the disease pathogenesis that mirror the molecular heterogeneity and anomalies found in human primary breast tumors. By analyzing the genomic and epigenomic profiles of both cancerous and non-cancerous breast tissues, especially primary breast tumors and pertinent breast cell lines [9,25,26], our data address knowledge gaps and provide novel insights into the aberrant interplay between epigenetic modulation and steroidogenic/estrogenic signaling, permitting a better understanding of breast pathogenesis, and pointing to improved therapeutic strategies for the management of BCs.

## 2. Results

### 2.1. Analyses of TCGA Pan-Cancer Normalized RNA-Seq Datasets for the Expression of Epigenetic Enzymes in Cancerous and Non-Cancerous Breast Tissues

Disruption of the homeostatic imbalance, involving epigenomic profiling, is a fundamental event in the pathophysiology of breast and other cancers [12,27,28]. As summarized in Table 1, 18 epigenetic enzymes are grouped into the following four classes: Class I (HDAC 1–3 and 8), Class IIa (HDAC 4, 5, 7 and 9), Class IIb (HDAC 6 and 10), Class III (SIRT1–7), and Class IV (HDAC 11). Genomic analyses of TCGA normalized pan-cancer BC datasets (https://gdc.cancer.gov/about-data/publications/pancanatlas, accessed on 1 May 2023) include 1095 breast tumors that possess both hormone-receptor-positive and -negative specimens and 139 non-cancerous breast tissues, and revealed marked alterations in most of these epigenetic enzymes in BCs in comparison to their non-cancerous counterparts (Table 1). Specifically, RNA-Seq data analyses indicated that the expression of 16 enzymes (excluding HDAC3 and SIRT5) was either significantly increased (HDAC1, 2, 8, 10, 11, and SIRT6 and 7) or decreased (HDAC4–7, –9, and SIRT1–4) in BCs. We reported that TCGA breast tumors are ~74% ER+, ~64% PR+, and ~51% Her2+, representing these tumors largely belong to luminal subtypes [10].

The magnitude of expression of these epigenetic enzymes was further visualized with a two-dimensional approach using TCGA RNA-Seq data cohorts that were utilized in Table 1. As illustrated by the heatmap, these enzymes demonstrated diverse expression profiles in breast tumors pertaining to luminal subtypes (Figure 1). It can be seen that robust expression was associated with HDAC1 and HDAC2 genes; in contrast, the lowest expression was visualized with SIRT4 and HDAC9. Other HDAC and SIRT members displayed varied expression levels. These findings indicate that the majority of these enzymes are either overexpressed or decreased in TCGA luminal BC subtypes, reinforcing the notion that alterations and/mutations (not assessed in this study) of these epigenetic regulators play crucial roles in the pathogenesis of BCs [12,28,29,30]. Dysregulation of various epigenetic enzymes in BCs raises the question of whether they are associated with cancer mortality.

### 2.2. Genomic Expression Profiling of Epigenetic Enzymes in TCGA Cancerous and Non-Cancerous Breast Tissues and Their Correlation with Overall Survival

To better understand the influence of aberrant regulation of epigenetic enzymes on the overall survival of BC patients, Kaplan–Meier curves were generated using TCGA RNA-Seq datasets involving higher (548 tumors) and lower (547 tumors) expression of those genes. The results presented in Figure 2 demonstrated that dysregulated expression of various epigenetic enzyme genes was not correlated, with the exception of HDAC2 (*p* = 0.03) and SIRT2 (*p* = 0.04), with the overall survival of patients afflicted with BCs. Notably, HDAC2 (Class I) and SIRT2 (Class III) were found to be increased and decreased in TCGA breast tumors, respectively. On the other hand, there was no significant relationship between overall survival rates and the dysregulation of other epigenetic enzyme genes: Class I, HDAC1 (*p* = 0.397), HDAC3 (*p* = 0.183), and HDAC8 (*p* = 0.783); Class IIa, HDAC4 (*p* = 0.891), HDAC5 (*p* = 0.935), HDAC7 (*p* = 0.802), and HDAC9 (*p* = 0.699); Class IIb, HDAC6 (*p* = 0.718) and HDAC2 (*p* = 0.781); Class III, SIRT1 (*p* = 0.932), SIRT3 (*p* = 0.065), SIRT4 (*p* = 0.426), SIRT5 (*p* = 0.156), SIRT6 (*p* = 0.604), SIRT7 (*p* = 0.503); and Class IV, HDAC11 (*p* = 0.175) (Figure 2). Clearly, while most of these epigenetic enzymes were dysregulated in BCs, their genomic expression levels did not correlate with the overall survival of BC patients. It is conceivable that an imbalance in epigenetic signaling presumably modulates diverse factors and hormone receptors that influence cholesterol trafficking and metabolism and promote estrogen/E2 accumulation, which is detrimental for the progression of breast carcinogenesis.

### 2.3. Analyses of a Variety of Steroidogenic Factors and Hormone Receptors Using TCGA Pan-Cancer Normalized BC RNA-Seq Datasets

In additional analyses, we studied the genomic expression levels of a total of 12 important factors including hormone receptor markers and steroidogenic enzymes, which play crucial roles in cholesterol availability for steroidogenesis and the progression of BCs [23,31,32]. To evaluate such relationships, TCGA breast tissues were categorized into three groups, i.e., Normal, ER+/PR+, and TNBCs, and the targeted factors chosen were based on their influence in regulating steroidogenesis, emphasizing estrogen and/or E2 biosynthesis [9,10,11,12,13]. As illustrated in boxplots (Figure 3), genomic analyses of the TCGA pan-cancer normalized BC datasets displayed diverse expression levels of these key factors, i.e., *LIPE* (**A**), *CYP27A1* (**B**), *STAR* (**C**), *STARD3* (**D**), *CYP11A1* (**E**), *HSD3B1* (**F**), *HSD17B1* (**G**), *CYP19A1* (**H**), *ESR1* (**I**), *ESR2* (**J**), *PGR* (**K**), and *ERBB2* (**L**). Genomic expression of these factors was found to be increased, decreased, or unaltered in TCGA BC subtypes compared with their non-cancerous counterparts, suggesting they differently influence estrogen-induced breast tumorigenesis.

### 2.4. Genomic Expression of Positive and Negative Regulatory Transcription Factors Using TCGA Cancerous and Non-Cancerous Breast RNA-Seq Datasets

The transcriptional machinery, involving STAR-mediated steroid biosynthesis, is coordinated by both enhancer elements that ‘switch on’ and silencer elements that ‘switch off’ gene expression [21,22,23,33,34]. To gain molecular insights into estrogen/E2-mediated breast carcinogenesis, the TCGA BC RNA-Seq datasets were evaluated for key trans-regulatory factors, including *CREB1, CREM, SF1, NR4A1, CEBPB, GATA1, SREBF1, SREBF2, SP1, FOS, JUN, NR0B1*, and *YY1*. As summarized in Table 2, these transcription factors were both upregulated and downregulated in TCGA luminal BC subtypes when compared with normal breast tissue. Consequently, we reported aberrantly higher expression of STAR, concomitant with E2 synthesis, in hormone-sensitive human primary breast tumors, pertinent BC cell lines, and transgenic mouse model spontaneous breast tumors [13,16]. These findings suggest that an inequity in the transcriptional machinery results in STAR-driven estrogen/E2 accumulation for triggering breast tumorigenesis.

### 2.5. Analyses of RNA-Seq Datasets for the Expression of Epigenetic Enzymes in a Variety of Human Cancerous and Non-Cancerous Breast Cell Lines

The genomic profiles of various epigenetic enzymes were further assessed in a total of 43 cancerous and non-cancerous human breast cell lines, i.e., 24-luminal, 10-basal A, 7-basal B, and 2-normal mammary epithelial cells [25]. The generation of heatmaps showed varied expression patterns of these epigenetic enzymes in the TCGA BC RNA-Seq data cohorts (Figure 4). The genomic expression of 17 of these enzymes (data not available for HDAC7), demonstrated diverse patterns, being noticeably high with HDAC 1, 3, SIRT 1, and SIRT 7, and lower with HDAC 9 and SIRT 4, reinforcing the notion that epigenetic regulators are commonly altered and/or mutated in BC cells [6,12,29,35]. Importantly, the majority of these enzymes are dysregulated in the TCGA luminal BC subtypes, implicating that homeostatic imbalance in epigenetic enzyme levels accelerates breast pathogenesis.

### 2.6. Genomic Expression of Key Steroidogenic Factors and Hormone Receptors in Cancerous and Non-Cancerous Breast Cell Lines

Abnormalities in breast cancer cell lines, involving oncogenic signaling and genomic heterogeneity, have been reported to correlate with primary breast tumors, suggesting the importance of cell line models for studying molecular events in disease pathophysiology [25,26]. The hypothesis that estrogen/E2-dependent BCs involve the malfunction of key steroidogenic factors (*LIPE, CYP27A STAR, STARD3, CYP11A1, CYP19A1, HSD3B1, HSD17B1,* and *CYP19A1*) and hormone receptors (*ESR1, ESR2, PGR*, and *ERBB2*), facilitating E2 accumulation, was assessed. As illustrated in the heatmap, the RNA-Seq datasets of cancerous and non-cancerous 43 cell lines, involving hormone-dependent and -independent BC categories, displayed varying expression levels of these factors impacting estrogen/E2 biosynthesis (Figure 5). Whereas a markedly higher expression was observed with *STARD3* and *ERBB2*, the lowest expression was associated with *CYP19A1* and *ESR2* levels, in various breast cell lines. Conversely, moderate to high expression levels were observed with *LIPE, STAR, ESR1,* and *HSD17B1*. Genomic expression was the highest with *STARD3* in many cancerous breast tumors cells lines, indicating the involvement of this late endosomal protein in cholesterol/E2-promoted breast tumorigenesis.

### 2.7. Ingenuity Pathway Analysis Using BC Microarray and RNA-Seq Datasets

To gain more insights into diverse pathways, networks, and biological functions, ingenuity pathway analysis (IPA) was performed using high-throughput gene expression profiles connecting epigenetic enzymes, cholesterol, and STAR (Figure 6). IPA data revealed that HDACs/SIRTs, cholesterol, and STAR-coupled BC subtypes are coordinately targeted by numerous mRNAs impacting the disease pathogenesis. Noteworthy, HDAC2 and SIRT2 (connected with poor survival of BC patients), APOE (apolipoprotein E), ABCG8 (ATP-binding cassette subfamily G member 8), LDLR (low-density lipoprotein receptor), HMGCR (3-hydroxy-3-methylglutaryl-CoA reductase), and PCSK9 (proprotein convertase subtilisin/kexin type 9), in addition to STAR, were found to be significantly associated with tumorigenesis in the breast tissue (Figure 6). Data are either unavailable or scanty for a number of HDACs. Despite the limitations, IPA predicted/identified a number of downstream effectors and new targets whose relevance to breast pathogenesis will be assessed in our future investigations.

## 3. Discussion

Epigenetic enzymes play pivotal roles in a wide variety of biological processes, including transcription, protein expression and function, subcellular localization, cell proliferation and differentiation, immune function, and metastasis [7,12,29,36]. In addition, these epigenetic regulators influence not only the acetylation of histones in nucleosomes but also a variety of non-histone substrates, including many proteins that are involved in tumorigenesis, angiogenesis, apoptosis, and cell invasion [12,37]. Dysregulation in the levels of these epigenetic factors results in homeostatic disparity in the molecular networks that govern diverse cellular and biological processes and modulates cancer etiology. Moreover, the imbalance in HATs and HDACs has been implicated in the abnormal expression of tumor suppressor genes and proteins involved in various cancers, including BC [38]. As mentioned above, the majority of BCs are stimulated by the estrogen/E2 that is produced by the aromatization of androgens on the part of aromatase in the cholesterol biosynthetic pathway. Studies have shown that cholesterol, especially its oxygenated derivative, 27-hydroxyxholesterol (27-HC), by interacting with ER and liver X receptors, provokes carcinogenesis in the breast tissue [12,39,40]. Our present findings extend these observations and allow a better understanding of breast pathogenesis by demonstrating genomic, epigenomic, and molecular analyses of TCGA normalized pan-cancer datasets and pertinent breast cell lines, and consequently point to novel insights into therapeutics. Moreover, development of an IPA identified new factors and targets associated with BCs involving epigenetic and steroidogenic networks. Interestingly, aberrant regulation of epigenetic signaling is progressively associated with anomalies in key steroidogenic factors and hormone receptors that harmoniously provide estrogen/E2 buildup for developing breast tumorigenesis.

An overwhelming amount of evidence indicates that the abnormality in gene expression is a crucial event in the progression of breast and other cancers. Nonetheless, multiple signaling pathways, affecting genomic and epigenomic dysregulation, protein synthesis, cell cycle progression, and apoptosis, modulate breast pathogenesis [11,41]. Pharmacological inhibition of various HDACs, against BC and other cancers, has been shown to display favorable outcomes, including cell cycle arrest and induced apoptosis, anti-proliferation, apoptosis, differentiation, anti-angiogenesis, and the drug resistance of cancer cells [12,18,28,29,30,42]. In accordance, technological advances have targeted therapeutic strategies with a number of HDACIs [20,43,44]. However, to achieve a treatment regimen, it is important to precisely identify HDAC and/or SIRT members that are overexpressed in breast tumors. The comprehensive analyses of TCGA BC RNA-Seq datasets identified elevated levels of seven epigenetic enzymes, namely the HDAC 1, 2, 8, 10, 11, SIRT 6, and 7 members. Consistent with this, the genomic levels of many epigenetic regulators, especially HDAC 1- 3, 5, 6, 11, SIRT 1, and 2, were markedly increased in 43 human malignant and non-malignant breast cell lines, reflecting scenarios ostensibly similar to primary breast tissues. Therefore, inhibition of these HDACs emerges as a potential therapeutic target for combating BCs. In line with these findings, we have demonstrated that the inhibition of a variety of HDAC and SIRT members, with SAHA (targets HDAC1 and 2), panobinostat (targets I, II and IV HDAC classes), entinostat (targets HDAC1 and 3), inhibitor IV (targets SIRT2), PCI-34051 (targets HDAC8), and romidepsin (targets HDAC1, 2, 4 and 6), at clinical and preclinical doses, suppresses STAR expression and E2 synthesis, in hormone-sensitive human MCF7 and mouse primary cultures of breast tumor epithelial cells [13,16,45]. It has been reported that treatment of SAHA/vorinostat in MCF7 cells downregulates ERα via the ubiquitin-mediated pathway and results in the inhibition of cell proliferation and induction of apoptosis [46,47]. Additionally, knockout of HDAC2 has been shown to inhibit cell proliferation, colony formation, migration, and cell cycle progression in TNBC cells and altered tumor growth in vivo [48]. Studies have reported that inhibition of Class I and Class II HDACs, especially 1, 2, 4, and 6, using trichostatin A and sodium butyrate affects cell proliferation on the part of mir-204 and ERα in MCF7 and MDA-MB-231 cells [49,50,51]. Recently, a combination of HDACIs (entinostat + vorinostat + belinostat), anti-HSP90 inhibitor (tanespimycin), and anti-helminthic inhibitor (niclosamide) has been shown to synergistically inhibit cell proliferation in TNBC and inflammatory BC cell lines [52]. In a clinical phase II trial with 43 patients with ER+/PR breast tumors, a combination of SAHA/vorinostat and tamoxifen (a selective ER modulator) demonstrated a ~50% reduction in tumors [53]. We recently reviewed the efficacy and specificity of a number of HDACIs in various clinical trials, either alone or in combination, for combating endocrine therapy resistance, along with favorable outcomes [12]. Therefore, it is plausible that HDACIs, by affecting multiple processes and signaling pathways, suppress/alter STAR-driven E2 accumulation, in both hormone-dependent and hormone-independent BCs.

The results of the present findings indicated that epigenetic dysregulation mirrored aberrant expression of factors that impact cholesterol trafficking/metabolism and steroid/E2 biosynthesis in both human primary tumors and pertinent cell lines. However, discrepancies in the gene expression profiles of certain factors, between primary tumors and cell lines, could be due to various ages, stages/grades, pathological features, and treatments of breast tumors for the former, and the age/stage-specific isolation of cells, passages, and culture conditions for the latter [9,25,26]. Interestingly, whereas the genomic levels of seven different epigenetic regulators were found to be markedly high in TCGA breast tumors, none other than HDAC2 were correlated with poor survival of BC patients. These findings suggest that aberrant expression of these epigenetic enzymes affects the steroidogenic machinery, facilitating estrogen/E2 assembly for promoting breast tumorigenesis. In pre-menopausal women, estrogens are essentially produced in the ovarian granulosa and placental corpus luteal cells; however, in peri- and post-menopausal women, extra-ovarian sites (adipose tissue, bone, skin, etc.), via paracrine and/or intracrine mechanisms, are key estrogenic sources that are liable for the progression of breast tumors in women ages 50 or over [12,54]. Despite diverse sources, estrogen/E2 is made from cholesterol, in which HSL/(*LIPE*), by catalyzing the hydrolysis of cholesterol esters, provides free cholesterol for steroidogenesis [23,32]. Nonetheless, higher levels of both *CYP27A1* and 27-HC are identified in human ER+ BCs, and have been reported to induce tumorigenesis [39,55,56]. Of importance, intramitochondrial transportation of cholesterol by STAR and the production of the first steroid, pregnenolone, by *CYP11A1* are indispensable events for the appropriate regulation of androgens and estrogens. It should be noted that STAR-governed cholesterol mobilization is the rate-liming step in steroid, as well as estrogen/E2, biosynthesis, and a fine-tuning in STAR’s transcriptional machinery has been shown to be mediated by a variety of transcription factors [21,22,57]. These cis-regulating factors bind to the sequence-specific DNA motifs present in the 5-flanking region of the STAR promoter, and the regulation of STAR gene transcription is coordinately influenced by both enhancer and silencer elements. Interestingly, however, these transcription factors were found to be drastically altered in TCGA breast tumors compared with their normal counterparts, which seemingly allows the modulation of STAR-driven E2 synthesis for promoting breast tumorigenesis. In support of this, we have uncovered that STAR is an acetylated protein, along with the identification of 15 lysine residues using LC-MS/MS, undergoing acetylation/deacetylation in ER+/PR+ BC cells, which enhances the expression and activity of STAR in optimal E2 synthesis [13]. However, the involvement of STARD3, with ~37% C-terminal homology to STAR having been shown to increase cholesterol biosynthesis in HER2+ BCs [58], in estrogen/E2-responsive breast tumorigenesis cannot be excluded. It is worth noting that STARD3 was initially cloned as a gene amplified in breast, gastric, and esophageal cancers [57,58], and the co-expression of both STARD3 and ERBB2 has been demonstrated in gastric cancer [59,60]. In the present study, RNA-Seq analyses of various steroidogenic factors revealed coordinate association of STARD3 and ERBB2 in a variety of cancerous and non-cancerous cell lines. It has been demonstrated that the expression of STARD3 is markedly higher in HER+ breast tumors in comparison with ER+/PR+ and TNBCs [61]. Studies have shown that the overexpression of STARD3 is associated with increased cholesterol biosynthesis in HER2+ breast cancer subtypes [58,62]. Considering differential expression and prognostic relevance, STARD3 has recently been proposed as a new biomarker in HER+ BCs [63]. Regardless of the factors involved, the E2 levels in tumors and/or the circulation of BC patients can be 30 times at a higher stage specifically than those seen in non-cancerous individuals [13,64].

The mechanism suppressing estrogen/E2 biosynthesis, in combating BCs, can be influenced by a variety of events, including hormonal, genetic, and reproductive factors [12,65,66,67]. It is well known that estrogen signaling is either activated or eliminated depending on the balance between ERα (*ESR1*) and ERβ (*ESR2*, considered a tumor suppressor) signaling, in which the growth and survival of BCs are influenced by the upregulation of ERα [13,68]. We observed that ER+/PR+ human breast tumors and/or pertinent cell lines revealed a higher genomic expression of ERα compared with normal breast tissue or TNBCs. Considering the overexpression of ER and the importance of aromatase in estrogen/E2 biosynthesis, AIs have been frequently used for BC treatment, especially for the hormone-sensitive category, in post-menopausal women. AI therapy involves the continued deprivation of estrogens, along with disruption of ER signaling, and results in the development of unwanted side effects, including osteoporosis, breast atrophy, depression, and reduced libido [69,70,71]. Despite considerable success with AIs, resistance to endocrine therapy is critical with increased tumor progression, the acquisition of malignant phenotypes, and poor prognosis, requiring an improved therapeutic approach to countering BCs. Our current data revealed that genomic, epigenomic, and hormone receptor/steroidogenic parameters in TCGA primary breast tumors were parallel with BC cell line models, permitting a better understanding of the diverse signaling pathways altered in various BC subtypes, in which the identification of new targets/factors using IPA plays an important role for the potential targeted therapeutics. Under these circumstances, we have reported that STAR expression is aberrantly high in ER+/PR+ human BC cell lines and mouse models of spontaneous breast tumors, in comparison with their non-cancerous counterparts [12,13,16], suggesting that STAR acts as a tumor promoter. Along these lines, our findings provide evidence that suppression of STAR using a number of both FDA-approved and clinical-phase trial HDACIs, resulted in a reduction in E2 synthesis, in human and mouse hormone-sensitive BC cells [13,16,45], underscoring the potential of this cholesterol transporter as a novel therapeutic target for the management of ER+/PR+ BC. Studies are currently underway on the functional relevance of STAR acetylation–deacetylation events in the regulation of estrogen/E2 biosynthesis toward an in-depth understanding of the molecular events involved between epigenetic dysregulation and steroidogenic signaling in cancerous and non-cancerous breast tissue.

## 4. Materials and Methods

### 4.1. Analyses of TCGA Pan-Cancer Normalized RNA-Seq BC Datasets for the Expression of Various Epigenetic Enzymes

The RNA-Seq datasets were downloaded from The Cancer Genome Atlas (TCGA; https://tcga-data.nci.nih.gov, accessed on 1 June 2023) database (GDC (cancer.gov)) [9]. TCGA, a landmark cancer genomics program, in coordination with the NCI’s Center for Cancer Genomics and the National Human Genome Research Institute, offers a comprehensive understanding of the molecular basis of various cancers and their diagnosis, prevention, and treatment, using genome sequencing and bioinformatics [9,24]. Specifically, TCGA pan-cancer normalized, batch-corrected, and platform-corrected RNA-Seq datasets including 139 normal breast tissues and 1095 primary breast tumors were downloaded (accessed on 12 May 2023) from the UCSC Xena (xenabrowser.net) browser [24]. RNA-Seq datasets from over 11,000 patient samples across 33 cancer types were aligned with the human genome, and the aligned data were sorted and indexed in BAM format (https://www.ncbi.nlm.nih.gov/pmc/articles/PMC2723002, accessed on 1 May 2023). After being assessed for data quality, low-quality samples were removed from further analyses. The raw read counts were then normalized to correct for library size and gene length differences between samples. The TCGA RNA-Seq datasets were then normalized using the upper quartile method, because it is less sensitive to outliers and batch effects. In addition, the ComBat software (SVA package, Version 3.48.0) was used to correct for batch effects on parametric and non-parametric empirical Bayes frameworks (https://www.ncbi.nlm.nih.gov/pmc/articles/PMC2723002). The clinical characteristics of all these cancerous and non-cancerous breast tissues were evaluated and downloaded from the UCSC Xena platform [24].

The eBayes function in the *limma* package was used to estimate the posterior probabilities of differential expression genes between primary tumors and normal tissues [72]. Specifically, the gene-specific variances and mean–variance relationships were estimated using the empirical Bayes method (voom: Precision weights unlock linear model analysis tools for RNA-Seq read counts-PubMed (nih.gov)). The posterior probabilities of differential expression were calculated by shrinking the estimated gene-specific variances toward a common value. In addition, the fold change was computed for solid breast tumors relative to its non-cancerous counterparts, and a *p*-value corrected for false discovery rate [73,74].

### 4.2. Generation of Kaplan–Meier Curves and Overall Survival Analyses

Kaplan–Meier (KM) survival analyses were used to compute the estimates of overall survival using primary breast tumor samples [10,12]. The log-rank test was used to perform comparisons of overall survival distributions between samples with higher and lower gene expressions, where median expression was used as the cutoff value. Overall survival was defined as the time elapsed from study enrollment to death, with those living censored at the time of last follow-up. Utilizing TCGA breast cancer RNA-Seq datasets, KM curves were generated with high and low (50:50) expression for all 18 epigenetic enzyme genes.

### 4.3. Analyses of RNA-Seq Datasets for the Expression of Various Steroidogenic Marker Genes, Hormone Receptors, and Transcription Factors in Cancerous and Non-Cancerous Breast Tissues

RNA-Seq datasets pertaining to both TCGA cancerous and non-cancerous human breast tissues (1095 primary tumors and 139 normal specimens) and pertinent 43 breast cell lines for the expression profiles of a variety of steroidogenic markers, hormone receptors, and transcription factors involved in steroidogenesis were downloaded (accessed on 12 May 2023) from the UCSC Xena browser [9,11,24]. The RNA-Seq datasets for both cancerous and non-cancerous breast cell lines included a total of 43 different types, i.e., 24-luminal, 10-basal A, 7-basal B, and 2-normal. All of these breast cell lines were based on ATCC revealing the specific characteristics and specifications of different subtypes were based on a previous study [25]. Genomic datasets for both TCGA BC and pertinent breast cell lines were categorized into ER (ER+ and ER−), PGR (PGR+ and PGR−, and Her2 (Her2+ and Her2−), as specified.

### 4.4. Generation of Heatmaps Using Transcriptome and RNA-Seq Data Pertaining to Various Breast Cell Lines and TCGA Tumors

Heatmaps were generated for visualizing the relative expression of a variety of genes (epigenetic enzymes, hormone receptors, and transcription factors) using the Complex Heatmap software (Version, 2.16.0) in R (The R Project for Statistical Computing (r-project.org, Version 4.2.2). The interconnections of a variety of genes involved in influencing the steroidogenic machinery were screened via data mining of the TCGA BC RNA-Seq transcriptome and relevant clinical datasets [9,24,25]. Similarly, the RNA-Seq data were obtained from various human cancerous and non-cancerous breast cell lines from the UCSC Xena browser [24,25]. A correlation heatmap was generated and analyzed for the enrichment of various genomic factors. The relative expression of core genes in both various breast cell lines and human primary tumors visually displays differences in the expression profiles between cancerous and non-cancerous breast tissues.

High-throughput BC microarray and RNA-Seq datasets were utilized to develop an IPA to better understand diverse signaling pathways, downstream effectors, and networks impacting breast pathogenesis. IPA was generated using the QIAGEN Inc. system (Venlo, The Netherlands), (https://www.qiagenbio-informatics.com/products/ingenuity-pathwayanalysis, accessed between 1 June 2023).

### 4.5. Boxplot Analyses of Key Steroidogenic Markers in Normal and Cancerous Breast Tissues Using TCGA Transcriptome and RNA-Seq Datasets

The Illumina HTSeq FPKM data, as well as the phenotype and cancer subtype data, were download from the UCSC Xena platform [9,24]. Next, all the samples were categorized into solid tissue Normal, ER/PR+, TNBC, and others, based on their ER, PGR, and Her2 status. Boxplots were generated with the Normal (139 samples), ER+/PR+ (615 samples), and TNBC (123 samples) samples involving various factors connected to cholesterol trafficking, metabolism, and steroidogenesis, in addition to hormone receptors, which included *LIPE, CYP27A1, STAR, STARD3, CYP11A1, HSD3B1, CYP19A1, HSD3B1, ESR1, ESR2, PGR*, and *ERBB2*.

### 4.6. Statistical Analysis

Descriptive statistics were used to describe the distributions of various data. Comparisons among groups were performed either using a two-sample *t*-test or one-way analysis of variance (ANOVA). Statistical analyses, where applicable, were performed using ANOVA. A *p*-value less than 0.05 was considered statistically significant. All analyses were performed using the R computing software (version 4.3.1; R Foundation for Statistical Computing, Vienna, Austria).

## 5. Limitations

We consider the following limitations of this work due to the heterogeneity in the TCGA BC stages/grades, demographics, ages, and pathological topographies; thus, the data should be cautiously interpreted. Breast samples were grouped into different categories based on detectable and allied features as available in the RNA-Seq datasets reflecting the targeted parameters, and we excluded those samples without clarity. As a consequence, breast specimens are varied in numbers for cancerous and non-cancerous groups, and the results obtained have been conscientiously discussed and/or analyzed. Moreover, the certainty displayed cannot be inferred as we evaluated genomic, epigenomic, and transcriptional profiles using a variety of bioinformatics tools, including heatmap and boxplot analyses. While analyzed algorithms are presented in the same order in both primary breast tumors and BC cell lines, readers’ conclusions could vary and may warrant additional investigations. These findings, however, provide novel insights into the dysregulation of epigenetic enzymes and their correlation with the steroidogenic machinery, especially estrogen/E2 biosynthesis, ratifying breast pathogenesis. Aberrant regulation of epigenomic and steroidogenic signaling was mirrored considerably in both human primary breast tumors and relevant BC cell lines, underscoring that the notion of uncertainty is limited. Studies with a larger number of samples, along with demographic information, and different technological settings would be beneficial for precise conclusions.

## 6. Conclusions

Analyses of dysregulated genomic, epigenomic, hormone receptors, and transcriptional factors, impacting breast tumorigenesis, differ in a subtype-specific manner, reflecting their survival, maintenance, and therapeutics. Accordingly, our IPA identified a number of new targets and processes/factors impacting BCs. Hormone-sensitive BCs express ERα and are predominantly activated by estrogen/E2, along with elevated genotoxic and oncogenic signaling, in which a myriad of processes and signaling plays permissible roles. Analyses of TCGA RNA-Seq datasets, involving cancerous and non-cancerous breast tissues, revealed disruption in the levels of the majority of epigenetic enzymes in luminal BC subtypes compared with their normal counterparts. Epigenetic enzymes regulate a variety of cellular processes, including chromatin remodeling and genomic stability through the dynamic process of the acetylation and deacetylation of core histones [12,29,35]. Interestingly, however, aberrant expression of most of these epigenetic regulators was not correlated with the overall survival of BC patients, suggesting they influence other factors/processes for stimulating the disease pathogenesis. Noteworthily, malfunction in the steroidogenic machinery, involving androgen and estrogen biosynthesis, has been implicated in the pathogenesis of hormone-dependent BCs [10,12,45,75]. Genomic profiling of TCGA BC datasets provides evidence that epigenetic dysregulation is closely linked with abnormalities of the factors and hormone receptors that critically influence various steps in the steroidogenic pathways, ranging from cholesterol trafficking/metabolism to estrogen/E2 accumulation, and these events that are culpable for breast carcinogenesis and eventual fatal consequences. The genomic and epigenomic heterogeneity observed in the human primary breast tumors was parallel with a variety of pertinent breast cell lines. As such, aberrations in molecular features, associated with primary breast tumors and relevant cell lines, facilitate a better understanding of the mechanisms involved in breast tumorigenesis and help develop the targeted therapies for countering one or more BC subtypes. Studies have shown that HDACIs have multiple targets on cancer cells, including BCs, and display favorable outcomes in numerous aspects, ranging from cell cycle arrest to the downregulation of estrogen/E2 levels [12,16,17,18,19,76]. Accordingly, we reported that a variety of FDA-approved HDACIs, at therapeutically relevant doses, by altering STAR acetylation patterns, suppress E2 accumulation in both ER+/PR+ MCF7 cells and mouse primary cultures of enriched breast tumor epithelial cells [13,16]. By analyzing RNA-Seq datasets, pertaining to both human primary breast tumors and pertinent cell lines [9,25,26], these data identify the molecular mechanisms connected with dysregulated epigenetic and steroidogenic machinery influencing breast pathogenesis, which are fundamental for a detailed understanding of diagnostic, preventive, and therapeutic strategies for BCs. Moreover, these analyses elucidate therapeutic perspectives with the inhibition of HDACs, emphasizing the relevance of more targeted HDACIs for the management of this devastating disease.

## Figures and Tables

**Figure 1 ijms-24-16488-f001:**
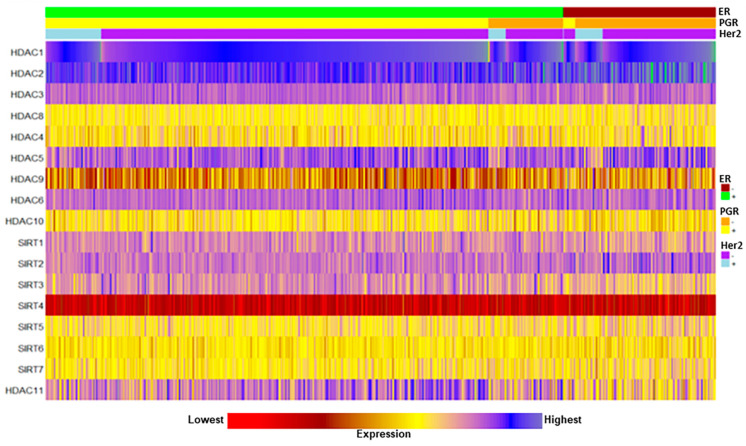
Generation of heatmap illustrating color-based graphical representation for genomic expression profiles of 18 different epigenetic enzymes (HDACs and SIRTs). TCGA pan-cancer normalized RNA-Seq datasets for cancerous (1095 tumors) and cancerous (139 specimens) breast tissues. ER, PGR (progesterone receptor), and Her2 categories are depicted on right side of the heatmap. These cancerous and non-cancerous breast tissues include ER+, ER−, PGR+, PGR−, Her2+, and Her2− subtypes. Bottom rectangular bar with ‘red’ and ‘blue’ colors at both ends represents the lowest and highest expression levels, respectively.

**Figure 2 ijms-24-16488-f002:**
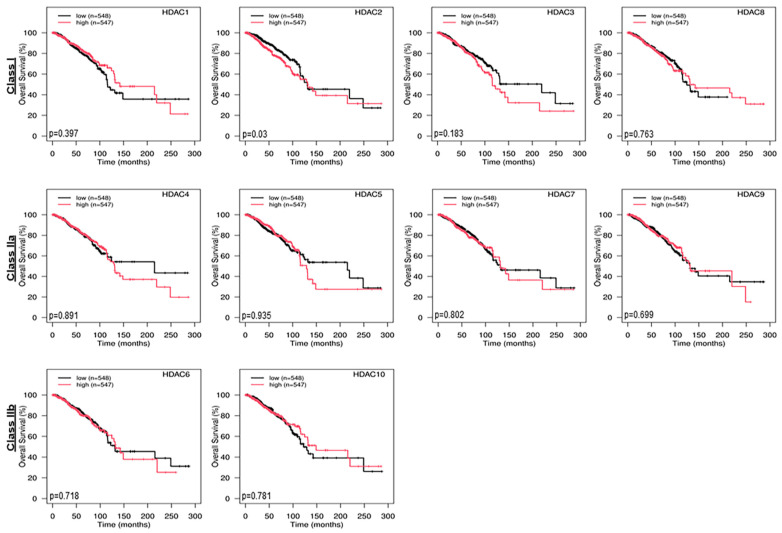
Kaplan–Meier (KM) plot analyses and determination of overall survival of BC patients in conjunction with diverse epigenetic enzyme expression levels. TCGA BC RNA-Seq datasets (1095 tumors) were used for generation of KM curves and their correlation with overall survival. KM survival curves were made with low (548 tumors) and high (547 tumors) expression for 18 different epigenetic enzyme (divided and shown in four different classes) genes, respectively. Black and red lines represent lower and higher expression of these genes, respectively. Note that KM survival curves were arranged according to the classes of these epigenetic enzyme genes. *p*-values for overall survival of each gene have been presented below these curves.

**Figure 3 ijms-24-16488-f003:**
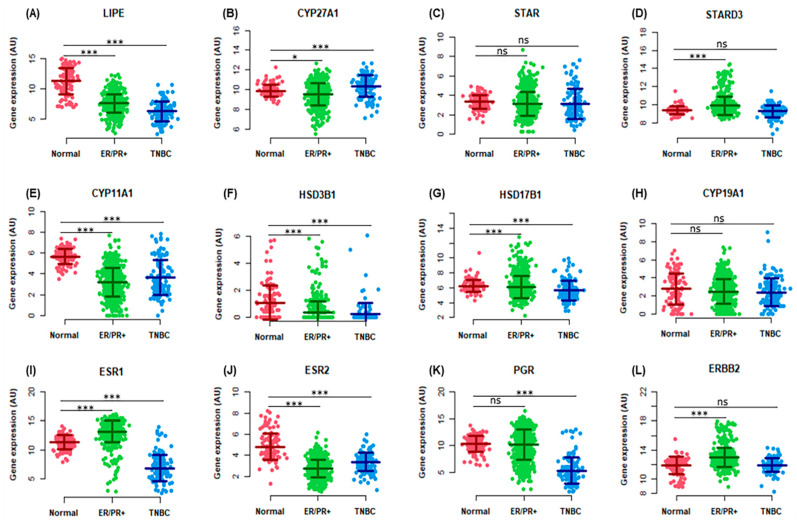
Boxplot analyses of a variety of steroidogenic factors and hormone receptors utilizing TCGA pan-cancer normalized RNA-Seq datasets. TCGA samples were divided into three different categories with varied samples numbers, i.e., Normal (dark orange; 39 samples), ER/PR+ (green, ER+/PR+; 615 samples), and TNBCs (blue; 123 samples). Steroidogenic factors and hormone receptors analyzed were the following: *LIPE* (**A**), *CYP27A1* (**B**), *STAR* (**C**), *STARD3* (**D**), *CYP11A1* (**E**), *HSD3B1* (**F**), *HSD17B1* (**G**), *CYP19A1* (**H**), *ESR1* (**I**), *ESR2* (**J**), *PGR* (**K**), and *ERBB2* (**L**). Note diverse expression levels of these steroidogenic factors and hormone receptors on Y-axes. *, *p* < 0.05; ***, *p* < 0.001; ns, not significant.

**Figure 4 ijms-24-16488-f004:**
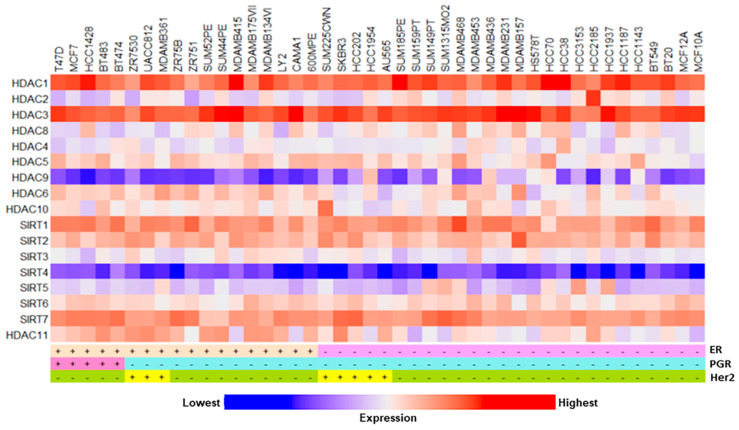
Generation of the heatmap illustrating color-based graphical representation for genomic expression of 18 different epigenetic enzymes (HDACs and SIRTs) in cancerous and non-cancerous breast cell lines. Breast cell lines analyzed are indicated on top on the heatmap (43 different lines). RNA-Seq datasets for these cell lines were downloaded from Xena browser and analyzed based on ER (+ and −), PGR (+ and −), and Her2+ (+ and −) subtypes, as indicated. Bottom rectangular bar with ‘blue’ and ‘red’ colors at both ends represents lowest and highest expression levels, respectively.

**Figure 5 ijms-24-16488-f005:**
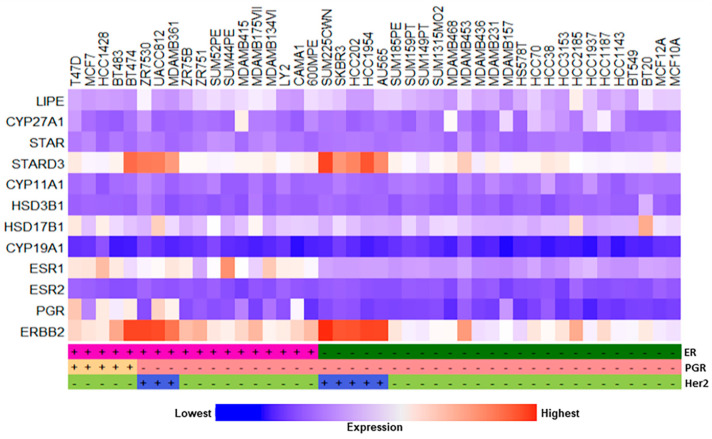
Genomic expression profiles of key steroidogenic factors and hormone receptors in a total of 43 cancerous and non-cancerous breast cell lines. Generation of heatmap illustrates color-based graphical representation of key steroidogenic factors and hormone receptor genes in various breast cell lines (mentioned on top on the heatmap). RNA-Seq datasets for these cell lines were downloaded from Xena browser and analyzed for *LIPE, CYP27A1, STAR, STARD3, CYP11A1, HSD3B1, HSD17B1, CYP19A1), ESR1, ESR2, PGR* and *ERBB2* genes. These cancerous and non-cancerous breast cell lines are ER (+ and −), PGR (+ and −), and Her2+ (+ and −) subtypes, as indicated. Bottom rectangular bar with ‘blue’ and ‘red’ colors at both ends represents lowest and highest expression levels, respectively.

**Figure 6 ijms-24-16488-f006:**
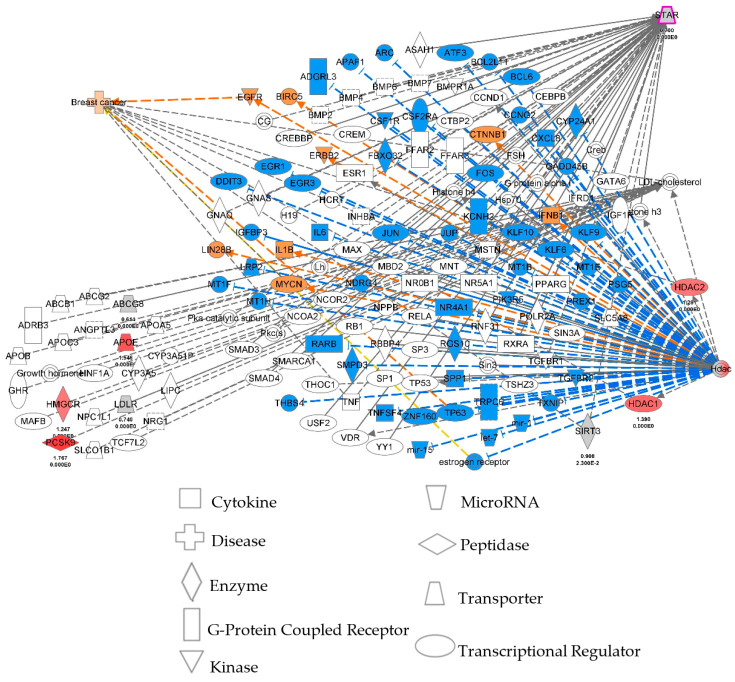
Ingenuity pathway analysis (IPA) illustrates epigenetic enzymes/HDACs, cholesterol, and STAR using high-throughput microarray and TCGA BC RNA-Seq datasets, and their connections to diverse gene expression networks and downstream effectors and targets. The blue lines connect biomarkers (in blue). We are interested in the symbols in oranges, which are affected by HDACs (orange lines). The symbols in red are those associated with lipid metabolism and cardiovascular health. All other markers (lines) are in black depicting inter-relationships among various signaling and/or molecular networks. Different nodes used are illustrated below IPA. Note that a number of HDACs are not illustrated in IPA due to the unavailability of relevant data.

**Table 1 ijms-24-16488-t001:** Analyses of TCGA pan-cancer normalized RNA-Seq datasets for expression of various epigenetic enzyme genes in cancerous and non-cancerous breast tissues.

HDAC Members	HDAC Classes	Expression in Normal Breast Tissues	Expression in Cancerous Breast Tissues	Fold Changes (95% Confident Interval)	*p*-Values
HDAC1	Class I	0.697	0.969	1.390 (1.298, 1.488)	<0.001, ***
HDAC2		0.913	1.169	1.281 (1.173, 1.399)	<0.001, ***
HDAC3		0.972	1.016	1.046 (0.996, 1.098)	0.071
HDAC8		0.833	1.037	1.245 (1.176, 1.317)	<0.001, ***
HDAC4	Class IIa	1.269	0.637	0.502 (0.457, 0.552)	<0.001, ***
HDAC5		1.260	0.883	0.701 (0.645, 0.762)	<0.001, ***
HDAC7		1.170	0.992	0.848 (0.789, 0.911)	<0.001, ***
HDAC9		1.115	0.681	0.611 (0.506, 0.738)	<0.001, ***
HDAC6	Class IIb	0.918	0.858	0.935 (0.883, 0.989)	0.022, *
HDAC10		0.476	0.683	1.434 (1.289, 1.596)	<0.001, ***
SIRT1	Class III	1.899	1.294	0.682 (0.630, 0.737)	<0.001, ***
SIRT2		0.967	0.731	0.755 (0.706, 0.808)	<0.001, ***
SIRT3		1.162	1.055	0.908 (0.838, 0.984)	0.022, *
SIRT4		1.329	0.951	0.716 (0.650, 0.788)	<0.001, ***
SIRT5		1.004	0.940	0.937 (0.875, 1.003)	0.063
SIRT6		0.389	0.717	1.846 (1.680, 2.028)	<0.001, ***
SIRT7		0.484	0.912	1.885 (1.720, 2.066)	<0.001, ***
HDAC11	Class IV	0.807	1.443	1.789 (1.570, 2.037)	<0.001, ***

*, *p* < 0.05; ***, *p* < 0.001.

**Table 2 ijms-24-16488-t002:** Analyses of TCGA BC pan-cancer normalized RNA-Seq datasets for expression of a variety of transcription factors.

Transcription Factors	Expression in Normal Breast Tissues	Expression in Cancerous Breast Tissues	Fold Changes (95% Confident Interval)	*p*-Values
CREB1	5.237	4.050	0.773 (0.728, 0.821)	<0.001, ***
CREM	0.902	0.903	1.001 (0.935, 1.072)	0.974
SF1	1.830	1.691	0.924 (0.892, 0.958)	<0.001, ***
NR4A1	2.224	0.524	0.235 (0.192, 0.289)	<0.001, ***
CEBPB	1.116	0.846	0.757 (0.661, 0.868)	<0.001, ***
GATA1	0.956	0.906	0.948 (0.837, 1.074)	0.427
SREBF1	3.391	6.327	1.866 (1.616, 2.155)	<0.001, ***
SREBF2	3.064	3.373	1.101 (1.004, 1.207)	0.050, *
SP1	2.047	1.620	0.791 (0.746, 0.838)	<0.001, ***
FOS	5.959	0.745	0.125 (0.099, 0.158)	<0.001, ***
JUN	4.286	1.431	0.334 (0.290, 0.385)	<0.001, ***
NR0B1	0.834	0.305	0.366 (0.301, 0.445)	<0.001, ***
YY1	2.032	2.295	1.129 (1.073, 1.189)	<0.001, ***

*, *p* < 0.05; ***, *p* < 0.001.

## Data Availability

Data in relation to the studies reported here are provided in this manuscript. However, the datasets analyzed during the current study are available from the corresponding author on reasonable request.

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
