# Peer review of "Epigenetic Dysregulation and Its Correlation with the Steroidogenic Machinery Impacting Breast Pathogenesis: Data Mining and Molecular Insights into Therapeutics"

_ijms, 2023, doi:10.3390/ijms242216488_

Round 1
Reviewer 1 Report
Comments and Suggestions for Authors
the present study is based on the data mining/data analysis. Authors analysed of The Cancer Genome Atlas (TCGA) pan-cancer normalized RNA-Seq 23 datasets and detected dysregulation of 16 epigenetic enzymes (increased HDAC1, 2, 8, 10, 11, and Sirtuins (SIRTs) 6 and 7; and decreased HDAC4-7, 9, and SIRT1-4 ). Analysis of the role of steroidogenic acute regulatory protein (STAR) is presented. The study is interesting and well-detailed.
There are only minor issues to address.
1. Graphical abstract can be useful for this study.
2. Figure 2 is tiny. It is impossible to read words. I had to enlarge figure 300% to be able to check what is written there.
3. The interesting observation – high co-expression of STARD3 and ERBB2 – was not discussed properly. Authors observed high expression of these, but did not discuss of the possible interaction between these factors. There is quite significant amount of published information about co-expression of these 2 genes ( see https://pubmed.ncbi.nlm.nih.gov/29190909/ ; https://pubmed.ncbi.nlm.nih.gov/24291029/ see thishttps://pubmed.ncbi.nlm.nih.gov/?term=STARD3+AND+ERBB2&sort=date).
4. Discussion section should be improved.
5. A hypothetical Signaling map can be presented to indicate connections between the detected factors.
Author Response
We would like to thank the reviewers for their valuable suggestions concerning our manuscript titled Epigenetic Dysregulation and Its Correlation to the Steroidogenic Machinery Impacting Breast Pathogenesis: Data Mining and Molecular Insights into Therapeutics. In light of the suggestions of the reviewers and the Editor, the manuscript has been appropriately revised with additional information. We believe that we have addressed and/or clarified all of the points raised, and sincerely hope that the revised version of manuscript meets the approval of reviewers and the editors for its consideration of publication. As suggested in the Editor’s letter, we responded point-by-point to each of the criticisms made by the reviewers. Our specific responses to the criticisms are as follows:
Comments and Suggestions for Authors:
Point 1: the present study is based on the data mining/data analysis. Authors analyzed of The Cancer Genome Atlas (TCGA) pan-cancer normalized RNA-Seq 23 datasets and detected dysregulation of 16 epigenetic enzymes (increased HDAC1, 2, 8, 10, 11, and Sirtuins (SIRTs) 6 and 7; and decreased HDAC4-7, 9, and SIRT1-4). Analysis of the role of steroidogenic acute regulatory protein (STAR) is presented. The study is interesting and well-detailed.
Response: We would like to thank the reviewer for her/his constructive commentaries on our manuscript, in addition to comment that the study is interesting and well-detailed.
There are only minor issues to address.
Point 2: Graphical abstract can be useful for this study.
Response 2: Considering in silico investigations involving genomic, epigenomic, transcriptional, and hormonal analyses pertaining to human breast cell lines and TCGA pan-cancer normalized RNA-Seq datasets in this manuscript, we felt reluctant to provide a graphical abstract to avoid clutter. However, additional data/information, especially ingenuity pathway analysis (IPA) that identified new targets and/or factors associated with BCs, have been included in the revised version, and the abstract has been updated accordingly.
Point 3: Figure 2 is tiny. It is impossible to read words. I had to enlarge figure 300% to be able to check what is written there.
Response 3: We agree with the reviewer that Figure 2 is tiny/small, as we presented KM plots for all classes, i.e., 18 epigenetic enzymes for comparisons. However, Figure 2 has been reorganized and enlarged for better visualization of the data presented and their effective understanding. The new Figure 2 has been presented in the place of the previous Figure 2.
Point 4: The interesting observation – high co-expression of STARD3 and ERBB2 – was not discussed properly. Authors observed high expression of these, but did not discuss of the possible interaction between these factors. There is quite significant amount of published information about co-expression of these 2 genes (see https://pubmed.ncbi.nlm.nih.gov/29190909/; https://pubmed.ncbi.nlm.nih.gov/24291029/; see this https://pubmed.ncbi.nlm.nih.gov/?term=STARD3+AND+ERBB2&sort=date)
Response 4: We thank this reviewer for her/his comments. We are aware of the involvement of STARD3 in BCs, especially HER2+ category, which was briefly discussed in the 3rd paragraph in the Discussion section. However, with regards to this reviewer’s suggestions, we have appropriately revised and/or expanded discussion section describing the possible interaction between STARD3 and EBRR2 with a number of relevant citations. These data have been discussed within the context of epigenetic dysregulation and the steroidogenic machinery and their correlation to breast pathogenesis.
Point 5: Discussion section should be improved.
Response 5: We have included new ingenuity pathway analysis (IPA) data in the revised version and have attempted to improve the discussion section with appropriate additional information (see Reviewer 3). We sincerely believe that discussion section has been improved in the revised version.
Point 6: A hypothetical Signaling map can be presented to indicate connections between the detected factors.
Response 6: A new Figure (Figure 6) associated with the IPA, involving signaling map and/or network, has been presented in the revised version. Specifically, IPA was performed to better understand diverse signaling pathways, networks, and biological functions in conjunction with epigenetic enzymes, cholesterol, and STAR, which utilizes high-throughput gene expression profiles and RNA-Seq datasets.
Reviewer 2 Report
Comments and Suggestions for Authors
Breast cancer stands as a significant global health concern characterized by diverse molecular subtypes and intricate regulatory networks. This study delves into the multifaceted interplay of dysregulated genomic, epigenomic, and transcriptional factors contributing to breast cancer development. Epigenetic enzymes like HDACs are key in chromatin remodeling and transcriptional control in breast cancer. Tailored treatments are needed due to distinct subtypes, especially estrogen-sensitive cases. HDAC inhibitors show potential in safely modulating estrogen production. Dysregulated steroidogenic pathways also complicate breast cancer.
Analyzing RNA-Seq data from The Cancer Genome Atlas (TCGA) provides invaluable insights into breast cancer subtypes and potential therapeutic avenues. Dysregulated expression of epigenetic enzymes is evident, particularly in luminal subtypes. However, these changes do not exhibit a straightforward correlation with the overall survival of patients, highlighting the complexity of breast cancer. While the study did not establish a direct connection between the expression of epigenetic enzymes and patient survival, it does enhance our understanding of breast cancer biology, potentially paving the way for further research and therapeutic advancements.
Therefore, I recommend this work for publication.
I have a few minor comments to address:
- Figure 2 has poor resolution, making it difficult to read gene names and p-values. Since many of the p-values are insignificant, it might be more suitable to include this as a supplemental figure.
- In Figure 3, it would be helpful to add a table for p-values or display them on the graph when comparing them with normal tissue.
Author Response
We would like to thank the reviewers for their valuable suggestions concerning our manuscript titled Epigenetic Dysregulation and Its Correlation to the Steroidogenic Machinery Impacting Breast Pathogenesis: Data Mining and Molecular Insights into Therapeutics. In light of the suggestions of the reviewers and the Editor, the manuscript has been appropriately revised with additional information. We believe that we have addressed and/or clarified all of the points raised, and sincerely hope that the revised version of manuscript meets the approval of reviewers and the editors for its consideration of publication. As suggested in the Editor’s letter, we responded point-by-point to each of the criticisms made by the reviewers. Our specific responses to the criticisms are as follows:
Comments and Suggestions for Authors:
Point 1: Breast cancer stands as a significant global health concern characterized by diverse molecular subtypes and intricate regulatory networks. This study delves into the multifaceted interplay of dysregulated genomic, epigenomic, and transcriptional factors contributing to breast cancer development. Epigenetic enzymes like HDACs are key in chromatin remodeling and transcriptional control in breast cancer. Tailored treatments are needed due to distinct subtypes, especially estrogen-sensitive cases. HDAC inhibitors show potential in safely modulating estrogen production. Dysregulated steroidogenic pathways also complicate breast cancer.
Analyzing RNA-Seq data from The Cancer Genome Atlas (TCGA) provides invaluable insights into breast cancer subtypes and potential therapeutic avenues. Dysregulated expression of epigenetic enzymes is evident, particularly in luminal subtypes. However, these changes do not exhibit a straightforward correlation with the overall survival of patients, highlighting the complexity of breast cancer. While the study did not establish a direct connection between the expression of epigenetic enzymes and patient survival, it does enhance our understanding of breast cancer biology, potentially paving the way for further research and therapeutic advancements.
Therefore, I recommend this work for publication.
Response: We would like to thank the reviewer for her/his constructive comments and valuable suggestions/insights on our manuscript. We agree with the reviewer regarding multifactorial and heterogeneous condition of breast cancers and their subtypes, involving many factors, pathways, signaling, and processes. However, this study essentially addresses some knowledge gaps and enhance our understanding of breast pathogenesis with reference to epigenetic dysregulation and its correlation to the steroidogenic machinery, in addition to potential new insights into therapeutics. Again, we thank this reviewer for recommending this work for publication.
I have a few minor comments to address:
Point 1: The Figure 2 has poor resolution, making it difficult to read gene names and p-values. Since many of the p-values are insignificant, it might be more suitable to include this as a supplemental figure.
Response 1: We agree with the reviewer regarding poor resolution of the Figure 2, which has been currently reorganized and enlarged so that gene names and p-values are easily visualized. The new Figure 2 has been presented in the place of the previous Figure 2. While the p-values for several HDACs are not significant with regards to overall survival of BC patients, the results provide novel insights indicating that dysregulation of epigenetic enzymes are not directly connected with patient mortality, inferring the involvement of additional factors for overall survival of patients afflicted with breast cancers. Therefore, we left this Figure in the main manuscript body in the revised version.
Point 3: In Figure 3, it would be helpful to add a table for p-values or display them on the graph when comparing them with normal tissue.
Response 3: We agree with the reviewer regarding addition of p-values in Figure 3. In the original submission, p-values were not provided in various panels of Figure 3, because of varied number of samples in three diverse categories. However, as suggested by the reviewer, we have added the p-values in relevant panels and have made comparisons with normal tissue, and the revised Figure has been replaced with the earlier Figure in the revised version.
Reviewer 3 Report
Comments and Suggestions for Authors
The authors have revealed the dysregulation of 16 epigenetic enzymes, as well as profiling steroidogenic factors and hormone receptor expression patterns along with their key trans-regulatory elements by analyzing TCGA pan-cancer normalized RNA-Seq 23 datasets and RNA-Seq datasets from different breast cancer cell lines. The datasets that have been used in this manuscript were being processed using different count files with different quantities and have not been shrunk or normalized through each other; therefore, the inter-cell line differential expression analyses cannot be fully trusted. The pathway analyses and the differentially expressed elements in them do not show any significance. Please normalize all datasets, shrink the fold changes considering other datasets and deep IPA analyses are recommended
They also provided general observations of the different expression levels of these epigenetic enzymes in TCGA breast tumours as well as their correlations to cancer mortality. They summarized that HDAC2 and SIRT2 were correlated with the overall survival of BC patients. The conclusion is weak as the expression levels of HDAC2 and SIRT2 vary during the entire surviving period. Patients with high or low HDAC2/SIRT2 expressions showed similar outcome. For further studies, combining the increase or decrease HDACs for multi-factor analysis and generating a model for predicting breast cancer patients’ survival.
In addition, the TCGA breast tissues were categorized into three groups, Normal, ER+/PR+, and TNBC, for further evaluating the relationships between hormone markers and regulators. Please explain what criteria are used to categorize them. There was a group of samples that were ER+/PR-. Please explain why the data from ER+/PR- was eliminated.
Figure 3 listed expression levels of the key factors in TCGA pan-cancer normalized BC datasets. However, most of the expression levels had no significant differences, especially STAR which was the research focus of the manuscript. The authors should add sections to provide validation to their data either using parallel datasets or in vitro experiments.
Overall, the manuscript was well written, however, the findings were not strong enough to support the conclusions. The authors should add sections to further explain the correlations between steroidogenic machinery and epigenetic dysregulation in breast cancer and illustrate how a variety of TFs function during this process, and provides robust network of epigenetics enzymes and steroidogenic machinery
Author Response
We would like to thank the reviewers for their valuable suggestions concerning our manuscript titled Epigenetic Dysregulation and Its Correlation to the Steroidogenic Machinery Impacting Breast Pathogenesis: Data Mining and Molecular Insights into Therapeutics. In light of the suggestions of the reviewers and the Editor, the manuscript has been appropriately revised with additional information. We believe that we have addressed and/or clarified all of the points raised, and sincerely hope that the revised version of manuscript meets the approval of reviewers and the editors for its consideration of publication. As suggested in the Editor’s letter, we responded point-by-point to each of the criticisms made by the reviewers. Our specific responses to the criticisms are as follows:
Comments and Suggestions for Authors:
Point 1: The authors have revealed the dysregulation of 16 epigenetic enzymes, as well as profiling steroidogenic factors and hormone receptor expression patterns along with their key trans-regulatory elements by analyzing TCGA pan-cancer normalized RNA-Seq 23 datasets and RNA-Seq datasets from different breast cancer cell lines. The datasets that have been used in this manuscript were being processed using different count files with different quantities and have not been shrunk or normalized through each other; therefore, the inter-cell line differential expression analyses cannot be fully trusted. The pathway analyses and the differentially expressed elements in them do not show any significance. Please normalize all datasets, shrink the fold changes considering other datasets and deep IPA analyses are recommended
Response 1: We have utilized TCGA pan-cancer normalized RNA-Seq datasets for analyzing a total of 18 members in the epigenetic histone deacetylase family and found that 16 members are dysregulated in BCs. Similarly, we have assessed genomic profiles of various factors in a total of 43 different non-cancerous and cancerous breast cell lines. In addition, we have performed deep IPA analyses that identified new downstream factors and targets (Figure 6), and this new Figure/data has been included in the revised version of manuscript with additional relevant information and/or discussion. Overall, this study provides additional insights into diverse signaling pathways/networks, and enhance our understanding of breast pathogenesis with reference to epigenetic dysregulation and its correlation to the steroidogenic machinery, in addition to potential therapeutics (see comments of Reviewer 2, as well as Reviewers 1 and 3).
Point 2: They also provided general observations of the different expression levels of these epigenetic enzymes in TCGA breast tumours as well as their correlations to cancer mortality. They summarized that HDAC2 and SIRT2 were correlated with the overall survival of BC patients. The conclusion is weak as the expression levels of HDAC2 and SIRT2 vary during the entire surviving period. Patients with high or low HDAC2/SIRT2 expressions showed similar outcome. For further studies, combining the increase or decrease HDACs for multi-factor analysis and generating a model for predicting breast cancer patients’ survival.
Response 2: We respectfully disagree with the reviewer, as equally divided patients’ samples (547 low and 548 high expressions, respectively) pertaining to TCGA pan-cancer normalized RNA-Seq datasets were utilized to generate KM plots for overall survival analyses, and the conclusions were made accordingly. Specifically, under similar experimental manipulations we found that both HDAC2 and SIRT2, but not other HDAC and SIRT members, were correlated with overall survival of BC patients. The utilization of TCGA BC RNA-Seq datasets, generation of KM plots, and overall survival of a number of other genes, have been previously analyzed and published in our earlier publications (Manna PR et al, 2022, Biomedicines, 10:1313; Cancers, 2019, 11, 423).
Point 3: In addition, the TCGA breast tissues were categorized into three groups, Normal, ER+/PR+, and TNBC, for further evaluating the relationships between hormone markers and regulators. Please explain what criteria are used to categorize them. There was a group of samples that were ER+/PR-. Please explain why the data from ER+/PR- was eliminated.
Response 3: Note that TCGA breast tissue samples were broadly and arbitrarily categorized into three different groups, involving Normal, hormone-dependent ER+PR+ (luminal subtype), and hormone-independent TNBC; accordingly, comparisons were made, in which we have included p-values in different panels in comparison with Normal (see Reviewer 2, point 3). We didn’t include a number of other categories, including ER+/PR- (luminal subtype), as there were either insufficient samples or overlapping nature of those categories.
Point 4: Figure 3 listed expression levels of the key factors in TCGA pan-cancer normalized BC datasets. However, most of the expression levels had no significant differences, especially STAR which was the research focus of the manuscript. The authors should add sections to provide validation to their data either using parallel datasets or in vitro experiments.
Response 4: The major theme of this study involves epigenetic dysregulation and its correlation to the steroidogenic machinery impacting breast pathogenesis and its potential therapeutics (title of this study), in which we have evaluated several key steroidogenic factors, including STAR, STARD3, and others (see Reviewer 1, point 4) that influence steroid biosynthesis. We have now added p-values in different panels in which ER+/PR+ and TNBC categories were compared with the Normal tissue (see Reviewer 2, point 3), and it has been included in the revised version.
Point 5: Overall, the manuscript was well written, however, the findings were not strong enough to support the conclusions. The authors should add sections to further explain the correlations between steroidogenic machinery and epigenetic dysregulation in breast cancer and illustrate how a variety of TFs function during this process, and provides robust network of epigenetics enzymes and steroidogenic machinery.
Response 5: We thank this reviewer for her/his comments on manuscript that it was well written. We sincerely believe that the results of the present study open up a new avenue in breast cancer research/field, address certain knowledge gaps, and enhance our understanding in a number of aspects, including epigenetic dysregulation and its correlation to steroidogenesis influencing the disease pathogenesis and insights into BC therapeutics. Importantly, this study also demonstrates that genomic, epigenomic, transcriptional, and hormonal anomalies that are qualitatively similar between human primary breast tumors and BC cell lines. Additionally, the inclusion of novel IPA data, involving robust network of epigenetics and steroidogenic factors connecting breast cancer, and appropriate relevant discussion, considerably improves the quality of this manuscript. Hope the reviewer will agree with us.
Reviewer 4 Report
Comments and Suggestions for Authors
The manuscript “Epigenetic Dysregulation and Its Correlation to the Steroidogenic Machinery Impacting Breast Pathogenesis- Data Mining and Molecular Insights into Therapeutics” reveals the dysregulation of 16 epigenetic enzymes in TCGA luminal breast cancer (BC) subtypes in comparison to their non-cancerous counterparts. Additionally, disruption of epigenetic signaling in BC subtypes was allied with genomic expression of various factors (i.e., LIPE, 33 CYP27A1, STAR, STARD3, CYP11A1, HSD3B1, HSD17B1, CYP19A1, ER, PGR, and ERBB2) instrumental for cholesterol trafficking/utilization in accelerating estrogen/E2 31 levels, in which steroidogenic acute regulatory protein STAR, influenced by a number of key transregulatory elements (i.e., CREB1, CREM, SF1, NR4A1, CEBPB, SREBF1, SREBF2, SP1, FOS, JUN, NR0B1, and YY1) mediates the rate-limiting step in steroid biosynthesis.
The entire manuscript is well written with sufficiently explained materials and methods section and a concise results section with adequate tabular and figure representation of obtained data. The discussion section is properly organized and accompanied by a corresponding and up-to-date reference list.
Except for Figure 1, which can be separated into 2 or 3 figures for better visualization of presented data, I do not have any special concerns or comments related to the manuscript.
Acceptance of the manuscript in its current form is suggested.
Author Response
We would like to thank the reviewers for their valuable suggestions concerning our manuscript titled Epigenetic Dysregulation and Its Correlation to the Steroidogenic Machinery Impacting Breast Pathogenesis: Data Mining and Molecular Insights into Therapeutics. In light of the suggestions of the reviewers and the Editor, the manuscript has been appropriately revised with additional information. We believe that we have addressed and/or clarified all of the points raised, and sincerely hope that the revised version of manuscript meets the approval of reviewers and the editors for its consideration of publication. As suggested in the Editor’s letter, we responded point-by-point to each of the criticisms made by the reviewers. Our specific responses to the criticisms are as follows:
Comments and Suggestions for Authors:
Point 1: The manuscript “Epigenetic Dysregulation and Its Correlation to the Steroidogenic Machinery Impacting Breast Pathogenesis- Data Mining and Molecular Insights into Therapeutics” reveals the dysregulation of 16 epigenetic enzymes in TCGA luminal breast cancer (BC) subtypes in comparison to their non-cancerous counterparts. Additionally, disruption of epigenetic signaling in BC subtypes was allied with genomic expression of various factors (i.e., LIPE, 33 CYP27A1, STAR, STARD3, CYP11A1, HSD3B1, HSD17B1, CYP19A1, ER, PGR, and ERBB2) instrumental for cholesterol trafficking/utilization in accelerating estrogen/E2 31 levels, in which steroidogenic acute regulatory protein STAR, influenced by a number of key transregulatory elements (i.e., CREB1, CREM, SF1, NR4A1, CEBPB, SREBF1, SREBF2, SP1, FOS, JUN, NR0B1, and YY1) mediates the rate-limiting step in steroid biosynthesis.
Response 1: We would like to thank the reviewer for her/his constructive comments in various aspects on our manuscript. To our knowledge this is the first study that correlates between epigenetic dysregulation and the steroidogenic machinery impacting breast pathogenesis, in addition to molecular insights into therapeutics.
Point 2: The entire manuscript is well written with sufficiently explained materials and methods section and a concise results section with adequate tabular and figure representation of obtained data. The discussion section is properly organized and accompanied by a corresponding and up-to-date reference list.
Response 2: We thankfully appreciate the comments of this reviewer on overall organization and appropriateness of our manuscript. Even so, we have included novel IPA data in the revised version, and different sections of the manuscript have been modified accordingly for further improvement and better understanding.
Point 3: Except for Figure 1, which can be separated into 2 or 3 figures for better visualization of presented data, I do not have any special concerns or comments related to the manuscript.
Response 3: We believe the reviewer meant for Figure 2, and not Figure 1, and we agree with the reviewer regarding small size of Figure 2. As mentioned above, the Figure 2 has been reorganized and enlarged for better visualization of the data presented and effective understanding. The revised Figure 2 replaced the previously submitted Figure.
Point 4: Acceptance of the manuscript in its current form is suggested.
Response 4: We sincerely thank the reviewer for recommending/suggesting the acceptance of our manuscript in its current form.
Round 2
Reviewer 3 Report
Comments and Suggestions for Authors
The authors provided reasonable answers to all the questions and revised the manuscript as suggested. Their observations conclude the landscape of epigenetic dysregulation in breast cancer patients and its correlation to the steroidogenic machinery and provide insights into potential therapeutic approaches involving inhibition of HDACs. I agree to accept this manuscript this time after minor revision. Please provide detailed legend for new Figure 6 including meanings of color lines and nodes, as well as Z-score.
Author Response
Comments and Suggestions for Authors:
The authors provided reasonable answers to all the questions and revised the manuscript as suggested. Their observations conclude the landscape of epigenetic dysregulation in breast cancer patients and its correlation to the steroidogenic machinery and provide insights into potential therapeutic approaches involving inhibition of HDACs. I agree to accept this manuscript this time after minor revision. Please provide detailed legend for new Figure 6 including meanings of color lines and nodes, as well as Z-score.
Response: We would like to thank this reviewer for her/his additional comments on our manuscript. As per the reviewer’s suggestions, we have revised and updated the manuscript, especially Figure 6 legend, with more detailed information, describing representations of color lines and nodes (that are presented immediately below Figure 6). We have also provided weblink information for the use of Qiagen’s IPA software/system for generation of the IPA (Materials and Methods and Acknowledgement sections, pages 16 and 18 in the revised version, respectively). However, we have not calculated Z-score, thus, not provided in the manuscript, as we are thoroughly and systematically analyzing diverse signaling, molecular networks, and biological functions, which require considerable amount of times for determining the precise Z-score. Even so, additional information provided in the Figure 6 legend, describing color lines, nodes, and their inter-relationships with diverse networks associated breast cancer, will certainly be helpful for better understanding of this newly generated IPA in more depth. Hope the reviewer and or Editor will agree with us.